

# Characterization of sympatric *Platanthera bifolia* and *Platanthera chlorantha* (Orchidaceae) populations with intermediate plants

Fabiana Esposito[1], Nicolas J. Vereecken[2], Maddalena Gammella[3], Rosita Rinaldi[3], Pascal Laurent[4] and Daniel Tyteca[1]

[1] Earth and Life Institute—Biodiversity Research Centre, Université Catholique de Louvain, Louvain-la-Neuve, Belgium
[2] Agroecology Lab, Brussels Bioengineering School, Université libre de Bruxelles (ULB), Brussels, Belgium
[3] Department of Biology, University of Naples Federico II, Napoli, Italy
[4] Unit of General Chemistry, Université Libre de Bruxelles, Brussels, Belgium

## ABSTRACT

*Platanthera bifolia* and *P. chlorantha* are terrestrial and rewarding orchids with a wide Eurasian distribution. Although genetically closely related, they exhibit significant morphological, phenological and ecological differences that maintain reproductive isolation between the species. However, where both species co-occur, individuals with intermediate phenotypic traits, often considered as hybrids, are frequently observed. Here, we combined neutral genetic markers (AFLPs), morphometrics and floral scent analysis (GC-MS) to investigate two mixed *Platanthera* populations where morphologically intermediate plants were found. Self-pollination experiments revealed a low level of autogamy and artificial crossings combined with assessments of fruit set and seed viability, showed compatibility between the two species. The results of the genetic analyses showed that morphologically intermediate plants had similar genetic patterns as the *P. bifolia* group. These results are corroborated also by floral scent analyses, which confirmed a strong similarity in floral scent composition between intermediate morphotypes and *P. bifolia*. Therefore, this study provided a much more detailed picture of the genetic structure of a sympatric zone between two closely allied species and supports the hypothesis that intermediate morphotypes in sympatry could reflect an adaptive evolution in response to local pollinator-mediated selection.

## INTRODUCTION

The evolution of reproductive isolation is a central topic in evolutionary biology. Flowering plants have evolved different ways to remain reproductively isolated from their congeners through various pre- and/or post-pollination barriers (*Coyne & Orr, 2004*; *Rieseberg & Willis, 2007*).

Orchids, a large and widespread family of flowering plants, are well known for their remarkable floral diversity. Ever since *Darwin (1862)*, orchid biology has focused on

Corresponding author
Daniel Tyteca,
daniel.tyteca@uclouvain.be

the fundamental causes of species richness and morphological diversity (*Cozzolino & Widmer, 2005*; *Schlüter & Schiestl, 2008*). Much of this diversity is associated with intricate relationships with pollinators, and has often been attributed to adaptation to specific pollinators (e.g., *Johnson, Linder & Steiner, 1998*)—an estimated 60% of all orchid species only have a handful of recorded pollinator species (*Tremblay, 1992*).

Pollinators act as a driving force in the reproduction and diversification of orchids (*Cozzolino & Widmer, 2005*) because they contribute to the establishment of reproductive isolation between species (*Van der Cingel, 1995*; *Cozzolino, D'Emerico & Widmer, 2004*; *Moccia, Widmer & Cozzolino, 2007*; *Stökl et al., 2008*; *Schatz et al., 2010*). Appropriate strategies for attracting pollinators and ensuring that cross-pollination is taking place efficiently are essential in the adaptation and evolution of the species. Particularly, orchids are known to have developed various and original strategies (reviewed by *Jersáková, Johnson & Kindlmann, 2006*). Given their strong influence on pollination efficiency, the adaptive value of floral traits displayed by orchids has received considerable attention from evolutionary biologists (e.g., *Edens-Meier & Bernhardt, 2014*).

Despite the presence of isolation barriers among species, natural hybridization is one of possible evolutionary processes that may occur in plants (*Stebbins, 1959*; *Arnold, 1992*; *Rieseberg, 1995*; *Abbott et al., 2013*). Considered as an important driving force in angiosperm diversification and speciation, this mechanism can originate "emergent" floral novelties between sympatric taxa (e.g., *Stebbins, 1959*; *Wissemann, 2007*; *Soltis & Soltis, 2009*; *Whitney et al., 2010*). The orchid family is known for having poorly developed genetic barriers to hybridization, even between genera (*Dafni & Ivri, 1979*; *Van der Cingel, 1995*; *Schatz et al., 2010*). Indeed, whenever genetically related taxa co-occur with an overlap in flowering periods and soil preferences, they may share pollinators and produce hybrids (e.g., *Cozzolino et al., 2006*).

*Platanthera* Rich., which belongs to subtribe *Orchidinae* (subfamily *Orchidoideae*), has apparently undergone an exceptional radiation in floral form and pollination syndrome (*Hapeman & Inoue, 1997*). The geographic distribution of *Platanthera* species—also known as "butterfly orchids"—covers most of the temperate zone throughout the Northern Hemisphere (*Hultén & Fries, 1986*) and this orchid genus encompasses five species in mainland Europe, two of which are widespread: *P. chlorantha* (Custer) Rchb. and *P. bifolia* (L.) Rich. (*Bateman et al., 2009*). They can be distinguished on the basis of the caudicle length and the distance between the viscidia, which seem to be the main discriminating factors between the two species (*Nilsson, 1983*; *Nilsson, 1985*). These two closely related species exhibit not only morphological differences, but also distinct ecological preferences (*P. chlorantha* favouring dry, calcareous grasslands, while *P. bifolia* will be typically found in slightly wet to humid meadows on acidic soil, although ecotypes growing on drier, calcareous soils are frequent). Additionally, several pre-pollination barriers have been established between the two species (*Nilsson, 1983*).

Plants exhibiting intermediate morphological characteristics in mixed populations of the two species have often been interpreted as hybrids (e.g., *Nilsson, 1985*; *Maad & Nilsson, 2004*; *Claessens & Kleynen, 2006*; *Bateman & Sexton, 2008*; *Bateman, James & Rudall, 2012*). Despite the large number of presumed hybrids recorded between the two *Platanthera* species

(*Bateman, 2005*; *Claessens, Gravendeel & Kleynen, 2008*), genetic analyses that directly compare putative hybrids with the sympatric parental species are rare (but see *Bateman, James & Rudall, 2012*). Studying the morphology and the genetic constellation of sympatric populations using molecular markers may provide an opportunity to identify hybridization between orchid species and help investigate the type and strength of reproductive isolation (*Martinsen et al., 2001*; *Lexer et al., 2005*; *Moccia, Widmer & Cozzolino, 2007*; *Cortis et al., 2009*). Recently, a study on some Western-European *Platanthera* populations composed almost exclusively of intermediate looking individuals, based on morphology and molecular markers, concluded that such individuals were not hybrids, but constitute an independent lineage, distinct from both widespread species (*Durka et al., 2017*), and described as a distinct species with the name *P. muelleri* (*Baum & Baum, 2017*).

The level of geitonogamy was observed to be higher in *P. bifolia* than in *P. chlorantha* because the latter has a pollinarium-bending mechanism that prevents deposition of the pollinia directly after removal (*Maad & Nilsson, 2004*; *Maad & Reinhammar, 2004*). This process may also affect the probability of hybrid formation (*Ishizaki, Abe & Ohara, 2013*). An allopatric *P. bifolia* population with a high degree (i.e., almost 60%) of self-pollination was found by *Brzosko (2003)*, although self-pollination in *Platanthera* species is considered generally rare (*Nilsson, 1983*; *Maad, 2002*).

In the genus *Platanthera*, floral scent plays a crucial role in guiding pollinators to the flowers (*Nilsson, 1983*; *Nilsson, 1985*; *Tollsten & Bergström, 1993*). A strong fragrance is emitted after dusk, when pollinators (nocturnal moths) are most active (*Nilsson, 1983*; *Nilsson, 1985*; *Tollsten & Bergström, 1993*; *Hapeman & Inoue, 1997*; *Plepys et al., 2002a*; *Plepys, Ibarra & Lofstedt, 2002b*). Floral fragrances of *Platanthera* have been classified into linaloolic, lilac, geraniolic and benzenoic chemotypes depending on the main class of compounds present in the blend (*Tollsten & Bergström, 1993*; *Plepys et al., 2002a*; *Plepys, Ibarra & Lofstedt, 2002b*). Lilac volatiles together with various benzenoids are strong attractants, being the most important compounds of floral scent in attracting moths (*Plepys et al., 2002a*; *Plepys, Ibarra & Lofstedt, 2002b*). Furthermore, a change in floral scent composition has been suggested by *Nilsson (1983)* and *Nilsson (1985))* to prevent effective cross-pollination between both species (*Nilsson, 1978*; *Tollsten & Bergström, 1993*), by acting as a reproductive barrier via ethological mechanisms. *Tollsten & Bergström (1993)* discovered that important inter-individual and inter-population variation in floral scent exists, and may act as an adaptation in order to attract a wider range of local pollinator species.

In this study, we investigated allopatric and two mixed populations of *Platanthera bifolia* and *P. chlorantha* in which morphologically intermediate individuals have been classified firstly as hybrids. In order to determine whether intermediates are indeed hybrids, or constitute an independent lineage, or represent mere variants of one of the two species, we performed a comparative analysis by comparing (i) the floral morphology (ii) the genetic profiles morphotypes, and (iii) the chemical characteristics of floral scents between allopatric, sympatric species and mixed populations. Furthermore, we investigated (iv) the reproductive success in each population by quantifying fruit set, (v) the genomic

compatibility of the two species by performing manual self and cross-pollination, and (vi) the presence and the strength of pre and post-pollination barriers (pre and post-zygotic).

## MATERIALS AND METHODS

### Study species and sampling sites

*Platanthera bifolia* and *P. chlorantha* are two terrestrial orchids with a wide Eurasian distribution. The flowering period in Central Europe of both species occurs between May and July, partly overlapping in areas of sympatry (*Delforge, 2006*). The inflorescences of *Platanthera* display 10–25 white, hermaphroditic flowers, which open sequentially acropetally and possess a long, slender nectariferous spur as a backward extension of the lip. The length of the spur varies geographically in North-western Europe, and it is also positively correlated to the proboscis length of local pollinators (*Darwin, 1862*; *Nilsson, 1985*; *Maad & Nilsson, 2004*; *Boberg & Ågren, 2009*; *Boberg et al., 2014*). Also, the distance between the viscidia is an adaptation to various organs on the pollinator's head (*Nilsson, 1983*).

Although the two species are very close genetically (*Bateman, James & Rudall, 2012*) and show the same diploid chromosome number ($2n = 42$) (*Nilsson, 1983*), a few floral traits, or combination thereof, allow separation of individuals into discrete groups matching the two species (*Darwin, 1862*; *Nilsson, 1978*; *Nilsson, 1983*; *Nilsson, 1985*). *P. bifolia* presents a small column with a narrow connective and anther pockets set almost parallel to each other, with a distance between the viscidia ranging between 0.2 and 1.1 mm, while the pollinium has a very short caudicle (0.2–0.5 mm); these characteristics imply that pollinaria will be attached to the pollinator's proboscis (Figs. 1A and 1B). The species is predominantly pollinated by hawkmoths (Sphingidae) (*Nilsson, 1983*).

By contrast, the column of *P. chlorantha* is wide, with a broad connective and the anther pockets set strongly divergent at the base. The pollinium has a relatively long caudicle (1.2–2.2 mm) and the distance between the viscidia is between 2.3 and 4.9 mm (Figs. 1E and 1F). This is considered to be an adaptation for attachment to the eyes of pollinators, which are mostly noctuids (Noctuidae) (*Nilsson, 1983*). In the intermediate plants the distance between the viscidia is, on average, larger than in *P. bifolia* and smaller than in *P. chlorantha* (1.3–2.3 mm), while the caudicle length lies between 0.6 and 1.2 mm (Figs. 1C and 1D). We based *a priori* identification of the individuals on these values intervals for the two specified characters (caudicle length and distance between the viscidia).

This intermediate form of the gynostemium may induce an inadequate attachment of pollinaria to the hairy labial palps of the moths (*Nilsson, 1978*). Therefore, the pollen of putative hybrids will often be lost because it will not reach the stigmas of other *Platanthera* individuals. As a result, crossing between hybrid derivatives seems poorly effective (*Nilsson, 1983*). This process should contribute to pre-pollination isolation and help to maintain the genetic integrity of each species (*Nilsson, 1983*; *Van der Cingel, 1995*; *Waser, 2001*; *Cozzolino, D'Emerico & Widmer, 2004*; *Scopece et al., 2007*).

We investigated sympatric populations with *P. bifolia*, *P. chlorantha* and intermediate morphotypes in two sites in Southern Belgium. As shown in Table 1, the two populations

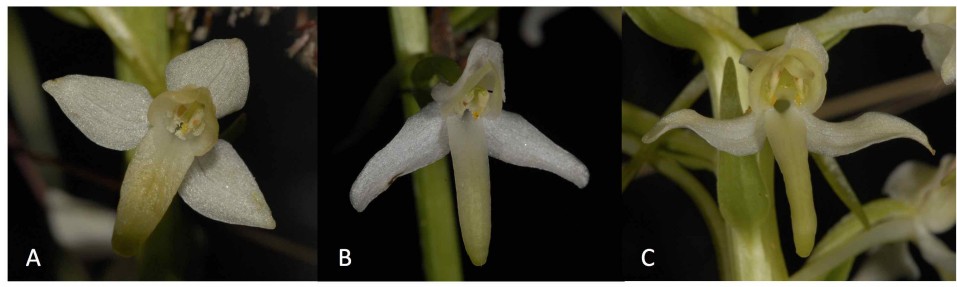

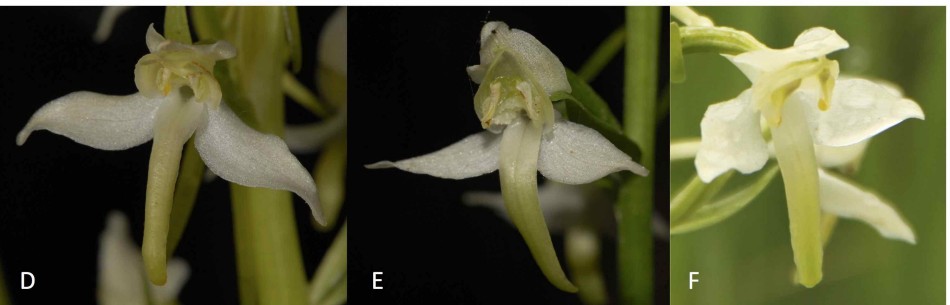

**Figure 1** **Pictures showing flowers of the plants investigated.** (A) *P. bifolia*, allopatric population, Navaugle, 8 July 2013. (B) *P. bifolia*, mixed population, Bois Niau, 3 June 2011. (C) *P. bifolia*, intermediate looking plant, Botton, 25 May 2011. (D) *P. bifolia*, intermediate looking plant, Botton, 25 May 2011. (E) *P. chlorantha*, mixed population, Bois Niau, 21 June 2010. (F) *P. chlorantha*, allopatric population, Transinne, 4 July 2013. Pictures deposited in the Herbarium of the Belgian National Botanic Garden (BR), Meise (all pictures D. Tyteca, except part (F): F. Esposito). See Appendix for correspondence.

**Table 1** **Numbers of plants submitted to the different kinds of analyses.**

| Station | Taxon | Morphology + fruit set | AFLP | Scent | Self-pollination | Cross pollination |
|---|---|---|---|---|---|---|
| Botton | *P. bifolia* | 30 | 20 | 10 | 6 | 15 |
| | *Intermediate* | 33 | 20 | 7 | 6 | |
| | *P. chlorantha* | 50 | 16 | 10 | | 15 |
| Bois Niau | *P. bifolia* | 20 | 20 | | | |
| | *Intermediate* | 18 | 18 | | | |
| | *P. chlorantha* | 14 | 14 | | | |
| Navaugle | *P. bifolia* | 36 | 20 | | 4 | 15 |
| Transinne | *P. chlorantha* | 41 | 20 | | 4 | 15 |
| Total | | 242 | 148 | 27 | 20 | 60 |

were sampled in the Calestienne region, one on a calcareous grassland (Tienne de Botton) and the other on a light birch–ash wood (Bois Niau). In addition, two allopatric populations were sampled: in the Famenne region (Navaugle) for *P. bifolia,* in a semi-wet meadow on acidic soil, and in the Ardenne region (Transinne) in a semi-wet neutral meadow for *P. chlorantha*. In the sympatric sites, plants were classified based on the values firstly suggested by Nilsson (1983), which will be used as the starting point for the investigations to

be conducted. In each site, plants showing good flowering conditions (i.e., fully flowering, with fresh flowers) were sampled randomly for the investigation. Selected individuals sampled for the morphological measurements were also subjected to genetic and chemical analyses (Table 1). Permissions to operate in the field in Nature Reserves were obtained from Walloon authorities in charge of Nature Protection (Département Nature et Forêts, Département de l'Étude du Milieu Naturel et Agricole). Photographic material of the studied populations was deposited in the herbarium of the Belgian National Botanic Garden (BR).

## Floral morphology and reproductive success: statistical analyses

In order to characterize the floral morphology of the different populations, four floral traits were measured: spur length (mm), caudicle length (mm), distance between the viscidia (mm) and labellum length (mm) (*Nilsson, 1983*; *Nilsson, 1985*; *Claessens & Kleynen, 2006*).

To test the null hypothesis of no morphological differences among taxa, we first conducted a non-parametric Multiple Response Permutation Procedure (MRPP) using VEGAN package version 2.0–5, with the average Bray–Curtis distances among samples weighted to group size and 999 random permutations (*Mielke & Berry, 2001*; *McCune & Grace, 2002*). Then, an analysis of similarities (ANOSIM) was performed using the average Bray–Curtis distances among samples and 1,000 permutations with the VEGAN package (version 2.0–5; *Oksanen et al., 2012*) in R (*R Core Team, 2015*) as an alternative way to test statistically whether or not there is a significant difference in morphological traits.

Furthermore, we carried out a multiple comparison with the Kruskal–Wallis test to evaluate the degree of association between samples and the Dunn's test (Dunn-Sidak-procedure) to determine which of the sample pairs are significantly different for each morphological trait.

In addition, we performed a canonical discriminant analysis using the morphological data. We applied a stepwise method with an *F* value of 3.84 to enter a variable, and *F* value of 2.71 to remove it (*Moccia, Widmer & Cozzolino, 2007*; *Jacquemyn et al., 2012a*). The discriminant function was derived using trait measurements from the two allopatric *Platanthera* populations. Then, we used the function to estimate the average floral morphology of each plant present in the sympatric zone (*Moccia, Widmer & Cozzolino, 2007*) that was used as morphological index. This analysis was conducted using the SPSS 21.0 statistical package (SPSS Inc., Chicago, IL, USA). We also performed a multivariate analysis (PCA) on correlation matrix, using the function prcomp, to summarize the information of morphological data. In addition, to compare fruit set (number of fruits/number of flowers) between *Platanthera* groups in both sympatric sites a multiple comparison with the Kruskal-Wallis test coupled with Dunn's test (Dunn-Sidak-procedure) was performed. These statistical analyses were performed in the software environment R version 3.2.1 (*R Core Team, 2015*).

## Manual crosses and pre and post-pollination isolation index

To determine the level of compatibility between species, experimental crosses were carried out in the sympatric area of Botton. Fresh flowers with intact pollinaria were randomly

selected. Interspecific hand-pollinations were performed by removing pollinaria through touching the viscidia with a plastic toothpick and placing them on the stigmas of plants of the other species. Crossing combinations were performed bi-directionally (*P. bifolia/P. chlorantha* and *P. chlorantha/P. bifolia*) with each plant providing and receiving pollen, and included control-treatments (Table 1).

To prevent the potential negative effects of over-pollination on fruit set and seed viability, a maximum of three flowers per individual were hand-pollinated. This experiment is based on *Xu et al. (2011)*. To prevent insect visits after experimental crossings, each inflorescence was covered with a pollination bag (to prevent pollination by insects) before and after the cross-pollination. Fruit initiation and development were monitored until fruits were mature (about one month after pollination). All crossed capsules were collected from the two investigated sympatric species and stored in silica gel. In addition, eight allopatric and 12 sympatric individuals of *P. bifolia* and intermediate morphotypes (Table 1) were also covered with a pollination bag before anthesis to determine the degree of autonomous self-pollination.

Seeds produced by interspecific (hand pollinations), intraspecific crosses and also in the autonomous self-pollination treatment were harvested and brought to the laboratory. Seeds were observed under a microscope (100× magnification) to distinguish seeds containing one large viable embryo from non-viable seeds (i.e., small or aborted embryos or no embryo). Samples of 300 seeds per fruit were scored in order to estimate the percentage of viable seeds for each fruit (*Xu et al., 2011*). The significance of different seed viability among interspecies and intraspecies crosses was assessed using Student's $t$-test, after normality testing of data distribution by the Shapiro test (*Royston, 1982*).

We also examined and quantified the effect of post-pollination barriers using indices of reproductive isolation (RI) (*Kay, 2006*). Based on the methods proposed in *Scopece et al. (2007)* and *Marques et al. (2014)*, we estimated two measures of post-pollination reproductive isolation. We firstly estimated the post-pollination pre-zygotic isolation index as the proportion of fruits formed after interspecific crosses in relation to the proportion of fruits formed after intraspecific crosses:

$$\text{RI post-pollination}_{\text{pre-zygotic}} = 1 - \frac{\text{average fruit set after interspecific crosses}}{\text{average fruit set after intraspecific crosses}}.$$

Then, we calculated post-zygotic isolation index as the percentage of viable seeds from interspecific crosses in relation to the proportion of viable seeds obtained from intraspecific crosses, describing the embryo mortality:

$$\text{RI post-pollination}_{\text{post-zygotic}} = 1 - \frac{\text{viable seeds formed after interspecific crosses}}{\text{viable seeds formed after intraspecific crosses}}.$$

In addition, since flowering time is known to contribute to the maintenance of phenotypic polymorphism, we estimated the strength of RI value, which corresponds to flowering phenology. The overall flowering period was recorded for both *Platanthera* species only at Botton site. Plants were checked every three days during one flowering season (2015). For the investigation of flowering phenology we examined: the beginning of blooming (first flower opened), the end of the flowering period (when the last flower opened). The RI

phenology index was calculated as: $RI_{phenology} = 1 -$ (overlapping flowering period between species (number of days)/flowering period (number of days)) (*Ma et al., 2016*).

## DNA extraction and AFLP analysis

In each population, a leaf fragment of ca. 2 cm$^2$ was collected for 10–20 plants of each of the taxa (see Table 1), and the plant tissue was desiccated using silica gel in individually sealed plastic bags. Genomic DNA was extracted using a slight modification of the CTAB protocol of *Doyle & Doyle (1987)*. Plant leaf material was macerated in 900 μL of standard CTAB buffer, incubated at 60 °C for 30 min, extracted twice with chloroform-isoamyl alcohol, precipitated with isopropanol and washed with 70% ethanol. Precipitated DNA was then resuspended in 30 μL of distilled water. We obtained AFLP fragments using the methods of *Vos et al. (1995)*, with modifications as reported in *Moccia, Widmer & Cozzolino (2007)* using fluorescent dye-labeled primers. Approximately 250 ng of genomic DNA was digested with *Eco*RI and *Mse*I restriction endonucleases, and then ligated with the appropriate adaptors. A pre-selective amplification of restriction fragments was conducted using a tem of 1 μL of restriction-ligation product and with *Eco*RIA + *Mse*IA or *Mse*IC as primers. After a preliminary screening for the variability and reproducibility, five selective combinations were chosen for this study: *Eco*RIA–MseICGG, *Eco*RIA–*Mse*IACT, *Eco*RIA–*Mse*ICCAA, *Eco*RIA–*Mse*ICGTA, *Eco*RIA–*Mse*IACTG.

The selective amplifications were conducted with 1 μL of a 1:10 dilution of pre-amplification product.

Separation and detection took place on a 3130 Genetic Analyzer (Applied Biosystems, Foster City, CA, USA). GeneScan-500 LIZ (Applied Biosystems) was used as IS (internal standard). The electrophoregram generated by the sequencer was analysed using the GeneMapper version 3.7 software package (2004; Applied Biosystems). Clear and unambiguous peaks, between 50 and 500 bp, were considered as AFLP markers and scored as present or absent in order to generate a binary data matrix. DNA of both allopatric species was amplified and run in duplicate to validate repeatability. The AFLP analysis was performed considering two data sets: the first, contained the Botton plants group + allopatric (plate-Bt), and the second contained the Bois Niau plants group + allopatric (plate-BN). These two data sets were run and scored independently.

We calculated FST values to estimate the population differentiation using the software AFLP-SURV v. 1.0 (*Vekemans, 2002*). Genetic structure was explored using Principal Coordinates Analysis (PCoA) in GENALEX (*Peakall & Smouse, 2006*). We performed a Bayesian clustering analysis that allows to estimate the number of genetic clusters (i.e., populations), allele frequencies within clusters, and the genetic composition of individuals, by assigning the latter to different groups in which deviations from Hardy–Weinberg equilibrium and linkage equilibrium are minimized (*Jacquemyn et al., 2012a*). Data were analysed in STRUCTURE v. 2.3.1 (*Pritchard, Stephens & Donnelly, 2000*; *Falush, Stephens & Pritchard, 2003*) assuming an admixture model and correlated allele frequencies with 50,000 burn-in steps and 100,000 MCMC (Markov chain Monte Carlo) steps and $K = 1$–10, with ten independent runs per $K$. The goal was to estimate the $K$ value that best fitted to our data.

The *K* value was assessed from the likelihood distribution (STRUCTURE output), which is the number of genetic clusters present in the data. *K* value fitting best with our data was selected using the Δ*K* statistic (*Evanno, Regnaut & Goudet, 2005*) produced by STRUCTURE HARVESTER (http://taylor0.biology.ucla.edu/struct_harvest/).

Finally, we used DISTRUCT (*Rosenberg, 2004*) to graphically display the output obtained with STRUCTURE.

NEWHYBRIDS (*Anderson, 2008*) was also performed to investigate the genetic profiles of the sympatric zone. We implemented a model that assumed two pure parental species and hybrids. This model assigns posterior probabilities for each individual to belong to one of the possibile six genotypic frequency classes: pure parental species, F1, F2, backcross to each parental species. A burn-in of 100,000 steps followed by run lengths of 1,000,000 steps was used (*Jacquemyn et al., 2012a*).

Moreover, the Hybrid index was estimated based in order to assess genome-wide admixture (*Buerkle, 2005*). This method calculated hybrid index (HI) based on a maximum likelihood and ranges between zero and one, corresponding to pure individuals of reference and alternative species, respectively. In our analyses, plants with a HI ranging between 0 and 0.2 were assigned to *P. bifolia*, whereas individuals with HI between 0.8 and 1 were assigned to *P. chlorantha*. We used AFLP data obtained from the allopatric *P. bifolia* and *P. chlorantha* individuals as parental data, while those obtained from the sympatric area were entered as putatively admixed individuals. This analysis was performed following the same parametric procedure proposed by *Jacquemyn et al. (2012a)*. The plot was produced with the mk.image function in INTROGRESS. The hybrid index was estimated to assess genome-wide admixture using the est.h function ((*Jacquemyn et al., 2012a*) incorporated in the R program INTROGRESS (*Gompert & Buerkle, 2010*). Finally, we correlated the molecular hybrid index with morphological index obtained with the discriminant function (described previously) using Spearman's rho method for non-normally distributed data (*Jacquemyn et al., 2012a*).

## Volatile collection and analyses of floral scents

In the sympatric zone in Botton, we sampled the volatile compounds emitted by flowers (the entire inflorescence) (Table 1). Floral scents emitted by the sympatric *Platanthera* species and the intermediate morphotypes were sampled for chemical analyses in the same phenological flower state, using a dynamic headspace adsorption technique during peak flowering time (June–July) and between 21:00 and 01:00 h local time, thereby matching the peak feeding times of most nocturnal moths (*Nilsson, 1978*). The same individuals were used to sample plant material for genetic analyses. The intact inflorescences were carefully enclosed in modified polyacetate bags (Pingvin frying bags, Art.nr 352: Kalle Nalo GmbH, Wiesbaden, Germany). The air, together with volatiles, was drawn through the bag by a battery-operated membrane pump, with a flow of 100 ml/min, into Teflon-PTFE cartridges containing 85 mg of the adsorbent Tenax-GR, mesh 60/80 (*Andersson et al., 2002*) for 60 min. Trapped scent compounds were eluted with 100 µL of cyclohexan and all samples were stored at −20 °C. Then, extracts were analysed by Gas Chromatography/Mass Spectrometry (GC-MS) on a Finnigan Trace Ultra GC coupled to a Finnigan POLARIS

**Table 2** Floral traits (Mean, with Standard Deviation) for *P. bifolia*, intermediate morphotypes and *P. chlorantha* for allopatric and sympatric populations.

| Morphology traits | Botton | | | Bois Niau | | | Allopatric pop. | |
|---|---|---|---|---|---|---|---|---|
| | *P. bifolia* | Interm. | *P. chlor.* | *P. bifolia* | Interm. | *P. chlor.* | *P. bifolia* | *P. chlor.* |
| | Mean | Mean | Mean | Mean | Mean | Mean | Mean | Mean |
| | SD | SD | SD | SD | SD | SD | SD | SD |
| Labellum length (mm) | 11.41 | 12.31 | 13.5 | 13.07 | 13.47 | 14.26 | 9.18 | 13.76 |
| | 1.8 | 1.32 | 1.49 | 1.39 | 1.23 | 0.66 | 2.45 | 0.28 |
| Spur length (mm) | 31.04 | 29.74 | 27.98 | 30.16 | 30.63 | 27.4 | 20.1 | 25.72 |
| | 3.83 | 3.4 | 2.99 | 2.88 | 2.18 | 0.71 | 2.64 | 0.43 |
| Caudicle length (mm) | 0.53 | 0.69 | 1.82 | 0.56 | 0.66 | 1.92 | 0.25 | 1.79 |
| | 0.11 | 0.12 | 0.16 | 0.1 | 0.16 | 0.07 | 0.26 | 0.03 |
| Viscidia distance (mm) | 0.96 | 1.48 | 3.61 | 0.64 | 1.39 | 3.91 | 0.3 | 3.9 |
| | 0.21 | 0.45 | 0.52 | 0.25 | 0.35 | 0.16 | 0.61 | 0.07 |

Q ion trap mass and equipped with a Restek RXI-5 MS column (30 m length × 0.25 mm diameter × 0.25 µm film thickness).

Aliquots of 1 µL of the extracts were injected in splitless mode first at 35 °C (4 min, followed by a programmed increase of oven temperature to 200 °C at a rate of 5 °C/min$^{-1}$) then at 200 °C for 1 min with an oven temperature to 270 °C at a rate of 10 °C/min. Helium was used as carrier. The proportional abundance of floral scent compounds (relative amounts with respect to aggregate peak areas, excluding contaminants) was calculated on the absolute amounts of compounds. Component peaks in the GC-MS chromatograms were quantified by integration of selected ion currents relative to one internal standard (IS) (2-phenylethanol, $C_8H_{10}O$). The Xcalibur$^{TM}$ Software was used and 2 µL of the internal standard was added for quantification of five samples of each group randomly chosen. Components were identified by their mass spectral patterns and chromatographic retention data (retention time and relative retention time). Furthermore, components were identified by comparing recorded mass spectra with the NIST08 and Wiley275 spectral databases with a probability of match >90%.

## RESULTS

### Morphology and fruit set

A preliminary MRPP analysis indicated that the floral morphology was significantly different between allopatric and sympatric groups in Botton ($A = 0.605$, $\delta_{obs} = 0.025$ $\delta_{exp} = 0.065$ $P < 0.001$) and Bois Niau ($A = 0.626$, $\delta_{obs} = 0.026 \delta_{exp} = 0.069$ $P < 0.001$). The ANOSIM analysis was in agreement with the results of the MRPP, since it also showed a significant difference between those groups (Botton: $R = 0.762$, $P < 0.001$; Bois Niau $R = 0.779$, $P < 0.001$). The results of the multiple comparisons with the Kruskal–Wallis and the Dunn's test displayed which trait differed between groups (reported as letters in Fig. 2; Table 2).

Intermediate morphotypes in both sites did not show any significant differences compared to sympatric *P. bifolia*, although values of morphological traits were slightly

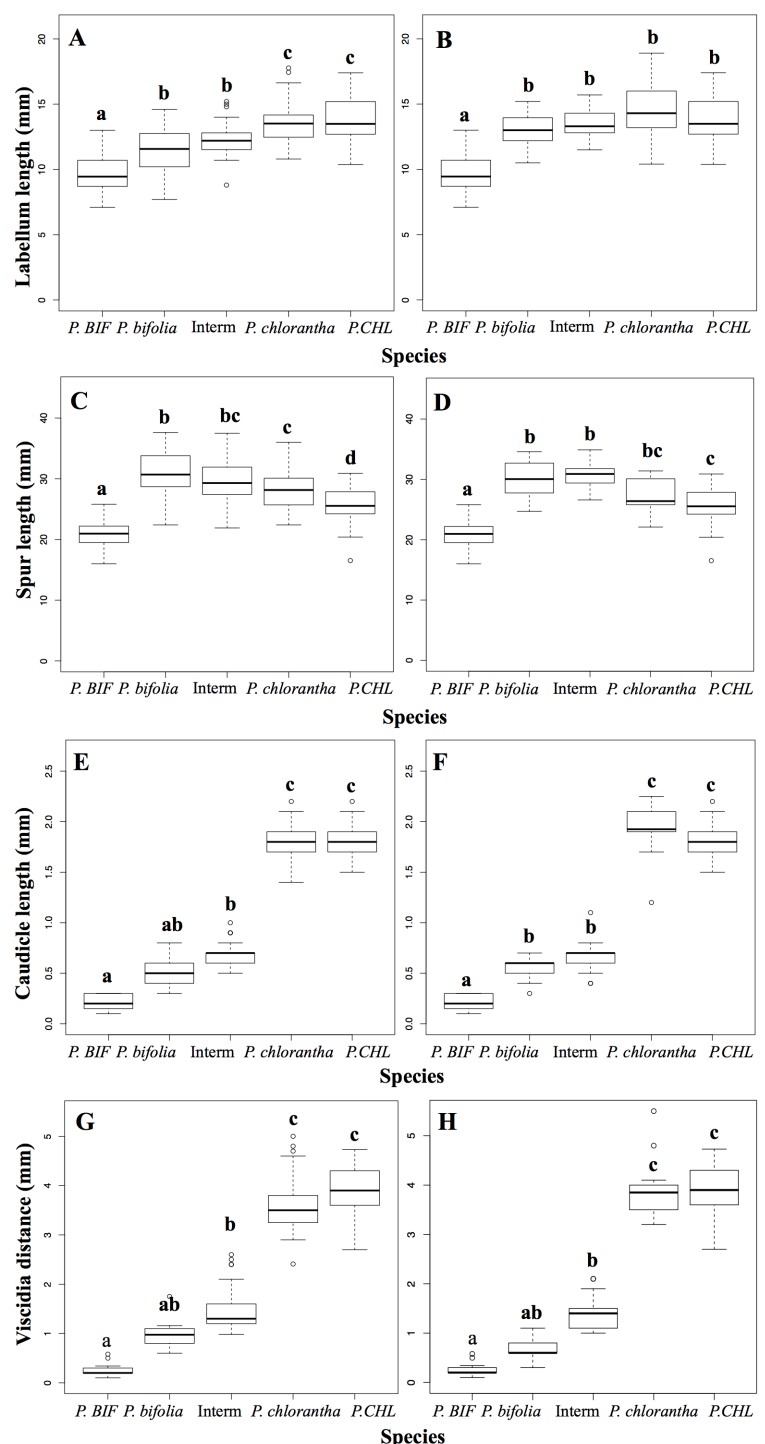

**Figure 2    Box plots of floral morphological traits among different taxa and populations.** (A, C, E, G): Box plots of allopatric populations and sympatric population of Botton. (B, D, F, H): Box plots of allopatric populations and sympatric population of Bois Niau. Different letters on top of boxplots indicate significant differences. Abbreviations: *P. BIF*, *P. bifolia* allopatric; *P. CHL*, *P. chlorantha* allopatric.

higher. Contrarily, comparing to allopatric *P. bifolia*, intermediate morphotypes showed significant differences for all morphological traits. On the other hand, intermediate morphotypes in Botton displayed significantly smaller values than those of sympatric *P. chlorantha,* with the exception of spur length; intermediate morphotypes in Bois Niau had significant differences only for viscidia distance and caudicle length.

Nonetheless, comparing intermediate morphotypes of both sites to allopatric *P. chlorantha*, all traits were significantly different, with the exception of labellum length for Bois Niau plants. Multiple comparisons (Kruskal Wallis and Dunn's test) did not show significant differences between intermediate morphotypes and sympatric *P. bifolia* for viscidia distance and caudicle length. The differences in these floral traits may be hidden by the extreme morphological values displayed by allopatric populations. We observed that caudicle length and viscidia distance in allopatric *P. bifolia* ranged from 0.1 to 0.3 mm and 0.1 to 0.58 mm, respectively, whereas the sympatric *P. bifolia* form had caudicle lengths varying between 0.3 and 0.9 mm and viscidia distances varying between 0.6 and 1.75 mm (results obtained considered both sites). It thus appears that those morphological characters are significantly different between allopatric and sympatric populations: these correspond to the two ecotypes as mentioned above (introduction).

The discriminant analysis using allopatric populations produced the function $D = 0.680$ (caudicle length), $+0.689$ (viscidia distance), (eigenvalue $= 57.119$; $\chi^2 = 150.112$ canonical correlation $= 0.991$; $P < 0.001$). Based on these functions, *P. bifolia* individuals received negative scores ($-7.36$) and *P. chlorantha* positive scores (7.36).

Principal components analysis of flower traits explained 89% of the variance along its two main axes between allopatric and Botton populations; the first principal component accounted for 63.31% of the species variation and had high positive loadings with viscidia distance, caudicle length and labellum length. The length of the spur was positively correlated with the second axis and the variation was 26%. The proportion of the explained variance between allopatric and Bois Niau populations was 92% (Fig. 3) and, the first principal component accounted for 62.97% of the variation and showed high positive loadings with viscidia distance, caudicle length and labellum length, whereas spur length accounted for 29% (Fig. 3) on the second axes.

Fruit set differed significantly between sympatric species and intermediate morphotypes in both sites (Botton: $\chi^2 = 12.34$ $P < 0.001$; Bois Niau: $\chi^2 = 9.07 P < 0.05$). Moreover, we observed a fruit-set advantage of *P. chlorantha* compared to *P. bifolia* in Botton (Dunn's test: $P < 0.01$) (Fig. 4A) and Bois Niau (Dunn's test: $P < 0.05$) (Fig. 4B), whereas a significant difference between *P. bifolia* and intermediate morphotypes was observed only in Bois Niau (Dunn's test: $P < 0.05$) (Fig. 4B).

## Manual crosses and pre and post-zygotic isolation index

The proportion of viable seeds obtained from interspecific crosses was not different between *P. bifolia* and *P. chlorantha* and between intraspecific crosses (allopatric populations). The average percentage of viable seeds was: 40% $\pm$ 33.8% SD for *P. bifolia* and 52.67 $\pm$ 37.38% SD for *P. chlorantha.* For intraspecific crosses we obtained for *P. bifolia* a mean of: 41.93% $\pm$ 25.68% SD, and for *P. chlorantha*: 38.07% $\pm$ 30.15% SD. The proportion

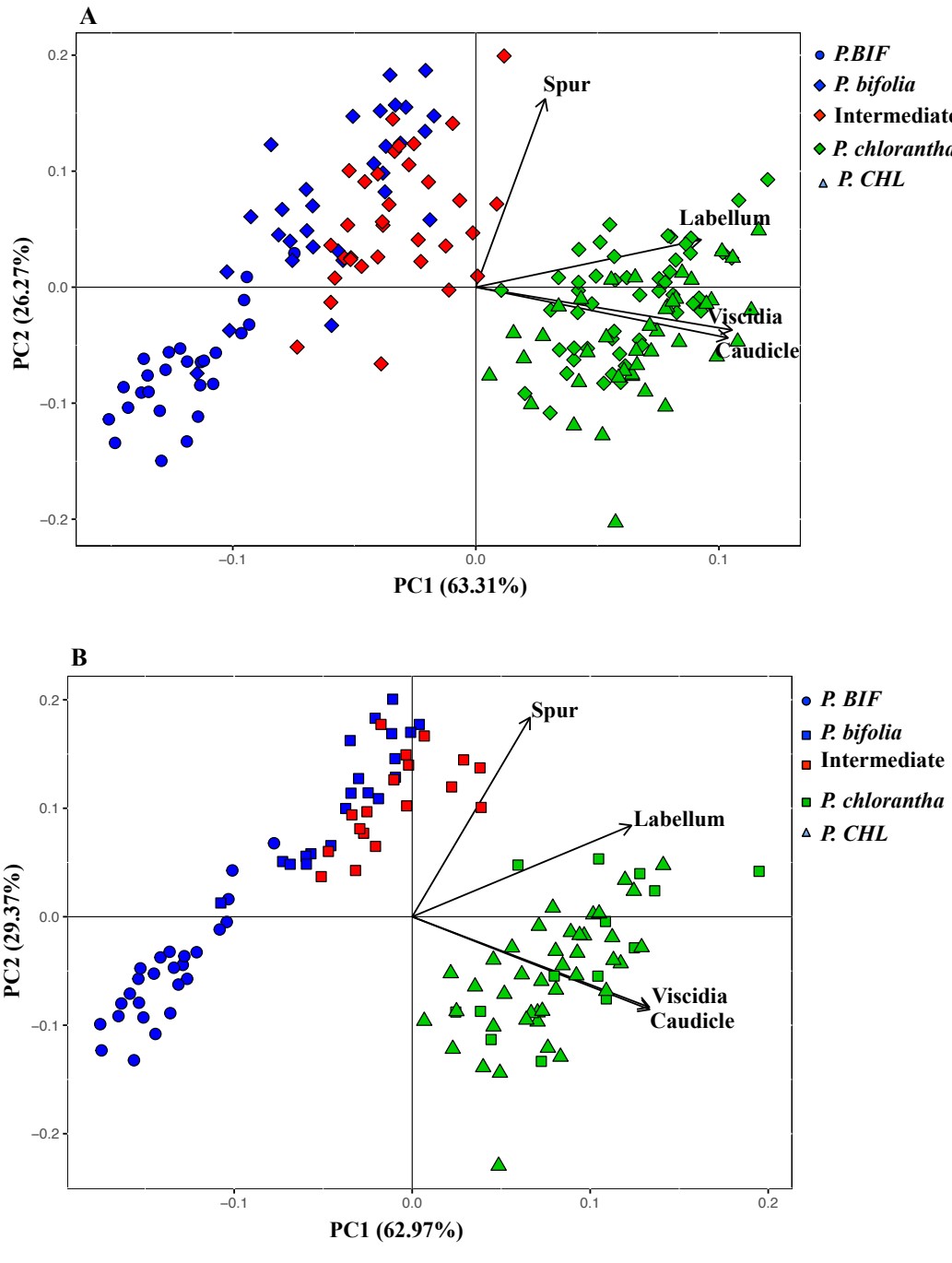

**Figure 3** **Principal component analysis (PCA) based on morphological flower characteristics of *Platanthera* allopatric taxa and sympatric population of Botton (A) and Bois Niau (B).** Floral characters represented in the PCA are: Spur, length of the spur; Labellum, length of the labellum; Viscidia, distance between the viscidia and Caudicle, length of the caudicle. *P. BIF*, *P. bifolia* allopatric; *P. CHL*, *P. chlorantha* allopatric.

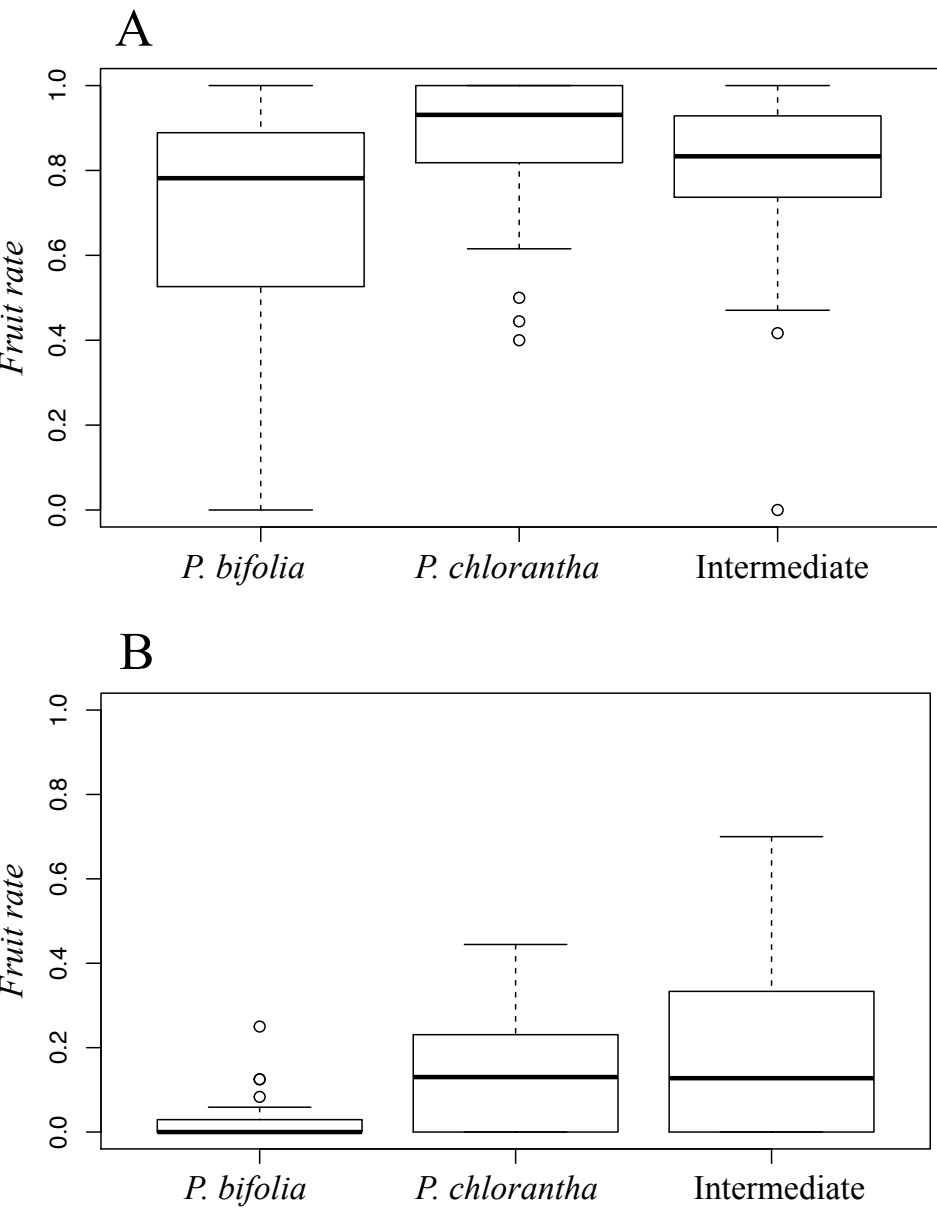

**Figure 4** **Fruit set in *P. bifolia*, intermediates and *P. chlorantha* of Botton (A) and Bois (B) Niau sympatric zones.** Bars indicate means and standard errors.

of viable seeds was not significantly different between interspecific crosses ($P = 0.56$) and intraspecific crosses ($P = 0.32$). For bi-directional crosses the post-pollination pre-zygotic isolation indices were negative (*P. bifolia*: $-0.43$; *P. chlorantha*: $-0.14$), which indicated that interspecies crosses performed better than intraspecies ones. Similarly, the post-pollination post-zygotic isolation indices were also weak, 0.22 for *P. bifolia*/*P. chlorantha,* and $-0.17$ for *P. chlorantha*/*P. bifolia.*

**Table 3** Genotype frequencies of species-specific markers in *P. bifolia*, *P. chlorantha* species and intermediate morphotypes of two sympatric zones.

| | Locus | *P. bifolia* | Intermediate | *P. chlorantha* |
|---|---|---|---|---|
| **Botton** | | | | |
| ACGfamCCAA | 61.55 | 0.00 | 0.00 | 1.00 |
| ATAnedCGTA | 65.2 | 0.00 | 0.27 | 1.00 |
| ATAnedCGTA | 81.56 | 0.00 | 0.00 | 1.00 |
| ATAnedCGTA | 105.72 | 1.00 | 0.77 | 0.00 |
| ACGfamCCAA | 117.45 | 1.00 | 0.77 | 0.00 |
| ACGfamCCAA | 316.75 | 0.00 | 0.00 | 1.00 |
| **Bois Niau** | | | | |
| ATAnedCGG | 70.85 | 0.00 | 0.00 | 1.00 |
| ATAnedCGG | 72.15 | 1.00 | 1.00 | 0.00 |
| ATAnedCGG | 77.22 | 0.00 | 0.00 | 1.00 |
| ATAnedCGG | 100.96 | 0.00 | 0.17 | 1.00 |
| ATAnedCGG | 106.95 | 1.00 | 0.66 | 0.00 |
| ATAnedCGG | 112.54 | 0.00 | 0.00 | 1.00 |
| AGGfamACT | 64.66 | 0.00 | 0.00 | 1.00 |
| ACGfamCCAA | 61.55 | 0.00 | 0.05 | 1.00 |

Self-pollination tests revealed a very low level of autonomous self-pollination in allopatric and sympatric populations analysed, since we observed just one flower forming a fruit on a sympatric individual of *P. bifolia* (out of 12 plants), but without viable seeds.

The overall period of flowering between *P. bifolia* and *P. chlorantha* largely coincided. For 21 days *P. bifolia* and *P. chlorantha* were flowering at the same time. More precisely, *P. bifolia* flowered from 4th of June to 29th of June, while the flowering period of *P. chlorantha* lasted from 22th of May to 24th of June. Therefore, *P. bifolia* flowered for 26 days and 34 for *P. chlorantha*. Therefore, $RI_{phenology} = 1 - (21/26) = 0.19$ for *P. bifolia*, and $RI_{phenology} = 1 - (21/34) = 0.38$ for *P. chlorantha*.

## Genetic diversity and differentiation

For the genetic analyses we excluded samples with a high number of missing values. Therefore, these analyses were carried out only on 28 allopatric individuals, 46 sympatric individuals in plate-Bt and 33 allopatric individuals, 43 sympatric individuals in plate-Bn. A total of 87 (plate-Bt) and 100 (plate-BN) polymorphic bands were detected in this study. The mean percentage of polymorphic loci was 79% for Botton and 76% for Bois Niau. We identified six and eight species-specific polymorphic bands in plate-Bt and plate-BN respectively; intermediate morphotypes showed the same polymorphism of *P. bifolia* for these species-specific sites (Table 3).

Pairwise $F_{ST}$ values were high between the intermediate morphotypes and *P. chlorantha* populations ($F_{ST} = 0.14$ plate-Bt and $F_{ST} = 0.16$ plate-Bn), but considerably lower between the intermediate population and *P. bifolia* ($F_{ST} = 0.01$ plate-Bt and $F_{ST} = 0.02$ plate-Bn).

In the principal coordinates analysis (PCoA) the first two axes explained 54% of variance for Botton and 57% of variance for Bois Niau population (Fig. 5). The PCoA

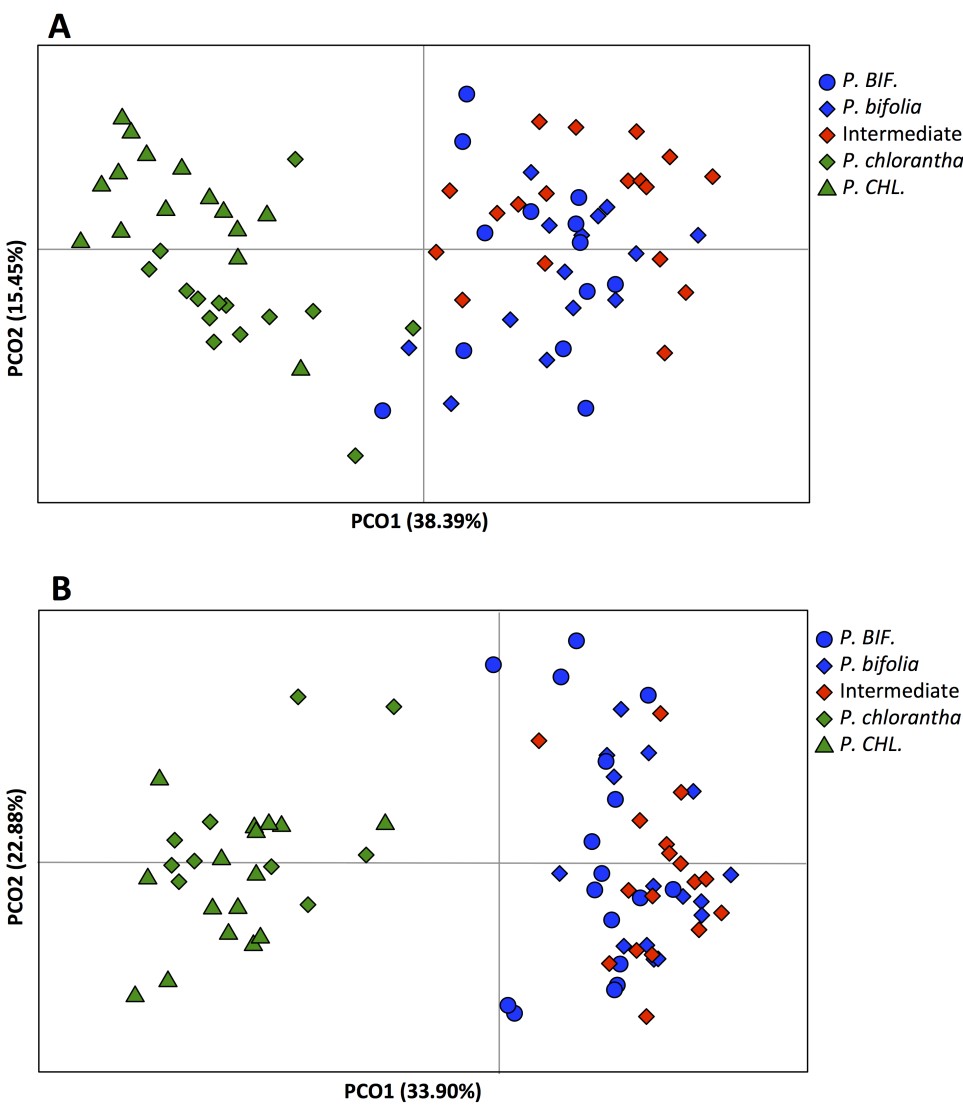

**Figure 5   Principal Coordinates Analyses (PCoA) based on AFLP of *Platanthera* allopatric taxa and sympatric population of Botton (A) and Bois Niau (B).** Abbreviations: *P. BIF*, *P. bifolia* allopatric and *P. CHL*, *P. chlorantha* allopatric.

clearly separated *P. bifolia* from *P. chlorantha* along the first axis, although there is a very slight overlap in the case of Botton plants. Moreover, PCoA plots for both plates identified two groups: (1) the intermediate morphotypes with allopatric and sympatric *P. bifolia*, and (2) the sympatric and allopatric *P. chlorantha* (Fig. 5).

The results obtained from the Bayesian admixture analyses with STRUCTURE (Fig. 6) showed that the likelihood $(\mathrm{Ln}P(D))$ increased greatly at $K = 2$ which, together with the fact that $\Delta K$ reached its maximum at $K = 2$, suggests the existence of only two genetic clusters for both plates (Fig. 6). Population clustering showed a consistent pattern indicating two independent genetic clusters: *P. bifolia* and the intermediate morphotypes formed a single cluster separated from *P. chlorantha* (Fig. 6).
**A**

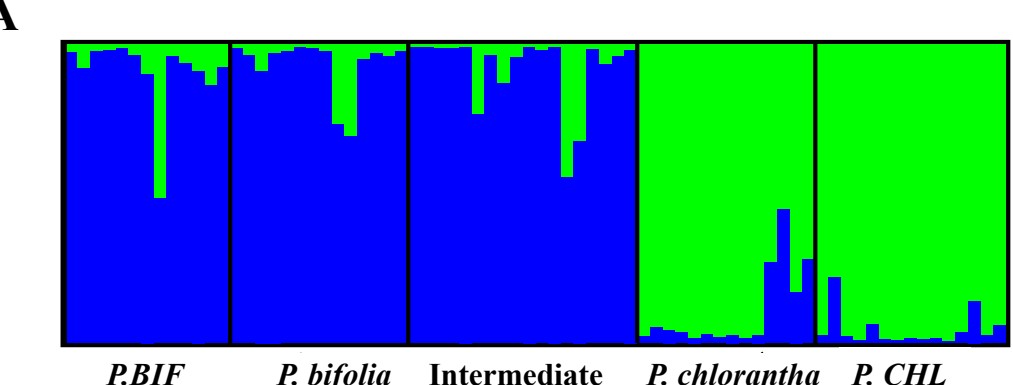

**B**

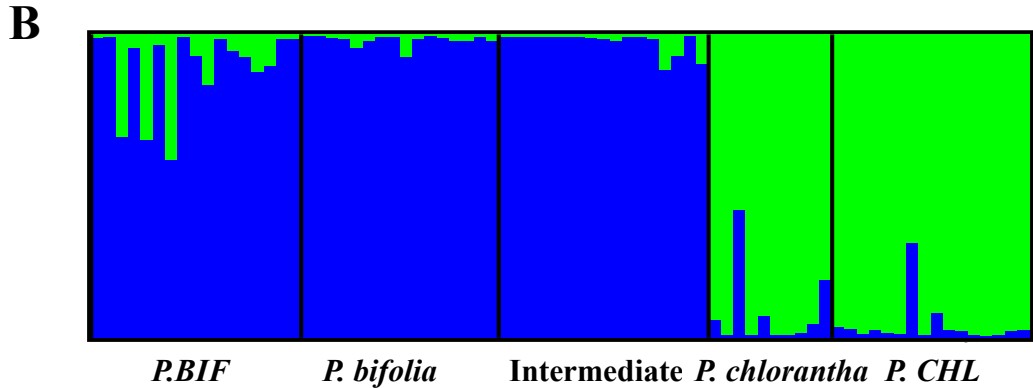

**Figure 6** **The structure of both sympatric populations inferred by Bayesian clustering is showed using STRUCTURE software ($K = 2$) with DESTRUCT output.** Columns represent individuals, while colours represent the proportion of their genome assigned to each of the two clusters. (A): Botton population + allopatric taxa, (B): Bois Niau population + allopatric taxa. Abbreviations: *P. BIF*, *P. bifolia* allopatric and *P. CHL*, *P. chlorantha* allopatric.

NEWHYBRIDS yielded similar results by assigning the intermediate morphotypes to the group of *P. bifolia* and revealing a low proportion of admixture genome in sympatric populations. Considering a threshold $q$-value of 0.9, we observed that 83% and 93% of individuals sampled in Botton and Bois Niau respectively were unequivocally assigned to *P. bifolia* and *P. chlorantha* and only four plants in Botton and one plants in Bois Niau that were identified as intermediate morphotype, had an admixed gene pool (Fig. 7).

In Bois Niau sympatric population all plants identified morphologically as *P. bifolia* and intermediate morphotypes had an hybrid index ranging between 0 and 0.2 supporting the previous results, whereas in Botton population most of them had an hybrid index ranging between 0 and 0.2 and only four individuals showed an hybrid index ranging between 0.3 and 0.4. Individuals firstly morphologically identified as *P. chlorantha* showed a hybrid index ranging between 0.4 and 0.7 in both sympatric populations.

Finally, molecular and morphological hybrid indices were significantly ($P < 0.001$) correlated in both sympatric sites (Fig. 8).
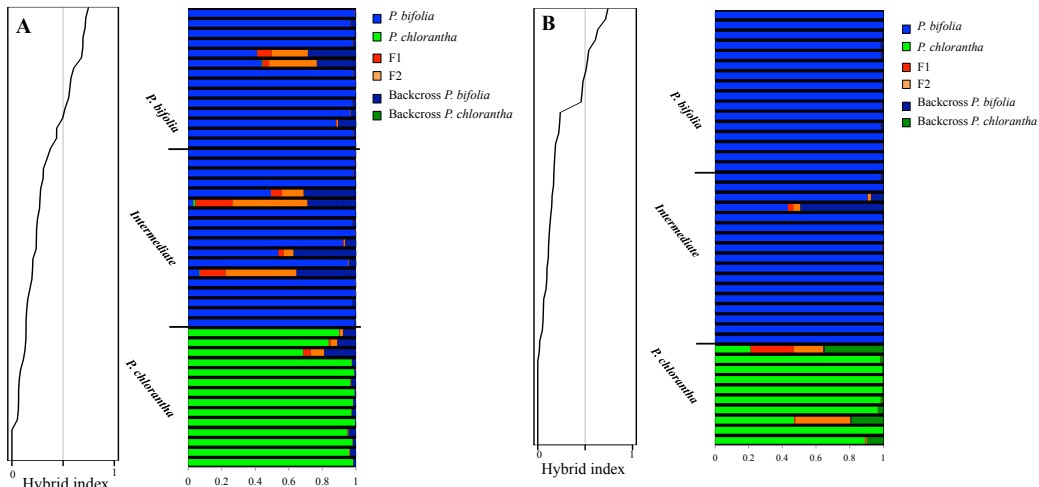

**Figure 7   Hybrid Index of sympatric populations (A: Botton; B: Bois Niau) and Bayesian inference of genotype class estimated with NEWHYBRIDS.** Hybrid Index of sympatric populations is showed on the left (A: Botton; B: Bois Niau). On the right, Bayesian inference of genotype class estimated with NEWHY-BRIDS. Colors represent the genotype classes and individuals are represented as rows. Within each row the extent of the component colors show the posterior probability of an individual with respect to each genotype class and the a priori group assignment is also displayed.

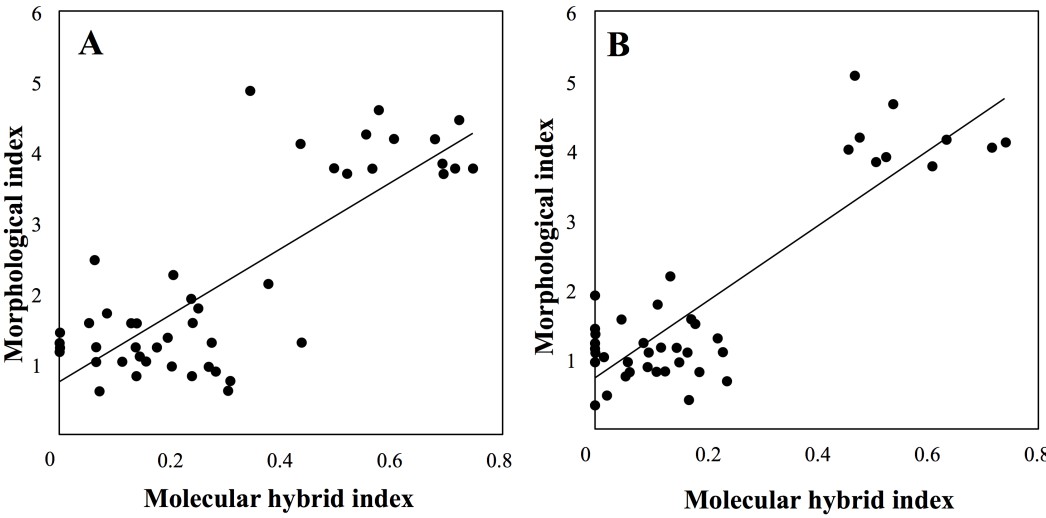

**Figure 8   Correlation between molecular and morphological hybrid indices in sympatric sites (A: Botton; B: Bois Niau).**

**Table 4   Floral scent profile (mean, standard deviation and occurrence) of bouquets emitted by the inflorescences of *P. bifolia* and *P. chlorantha* and the intermediate morphotypes.**  The table shows the relative amounts (in %) of odour compounds in headspace fractions of the different taxa.

| Volatile compounds (%) | *P. bifolia* Mean SD Occurrence | Intermediate Mean SD Occurrence | *P. chlorantha* Mean SD Occurrence |
|---|---|---|---|
| Total number of individuals | 10 | 7 | 10 |
| 3,7-Dimethyl-1,3,6-octatriene | 30.39 | 13.81 | 11.34 |
|  | 8.96 | 18.22 | 20.14 |
|  | 10 | 3 | 4 |
| 1,2-Hexanediol-2-benzoate | 23.5 | 39.92 | 0 |
|  | 15.05 | 31.02 | 0 |
|  | 10 | 7 | 0 |
| Santolinatriene | 36.1 | 33.5 | 0.03 |
|  | 19.45 | 26.88 | 0.09 |
|  | 9 | 5 | 3 |
| 3,7-Dimethyl-2,6-octadien-1-ol | 9.76 | 12.25 | 0 |
|  | 5.66 | 6.15 | 0 |
|  | 10 | 7 | 0 |
| 3-Carene | 0 | 0 | 13.16 |
|  | 0 | 0 | 15.61 |
|  | 0 | 0 | 9 |
| Benzyl acetate | 0.17 | 0.26 | 0 |
|  | 0.54 | 0.14 | 0.01 |
|  | 1 | 6 | 1 |
| 3,7-Dimethyl-2,6-octadien-1-ol acetate | 0 | 0 | 1.67 |
|  | 0.54 | 0.07 | 2.63 |
|  | 0 | 0 | 6 |
| Lilac aldehyde | 0 | 0.13 | 69.91 |
|  | 0 | 0.29 | 15.63 |
|  | 0 | 2 | 10 |
| Lilac alcohol | 0.08 | 0.14 | 3.88 |
|  | 0.16 | 0.16 | 2.64 |
|  | 3 | 4 | 10 |

## Floral scents

Based on the group of compounds that dominated a scent profile, the bouquets emitted by the inflorescences of *P. bifolia*, *P. chlorantha* and the intermediate morphotypes included a total of ten volatile compounds: two benzenoids and eight monoterpenoids (Table 4).

The following classes could be distinguished: lilac aldehydes, alcohol compounds, geraniolic compounds, and benzenoid compounds. Compared with *P. bifolia*, the scent patterns within *P. chlorantha* populations were less variable. Nevertheless, individuals of the two species showed a divergent chemical pattern. Specifically, the mean of relative percentage of lilac aldehyde in *P. chlorantha* was higher compared with the sympatric

*P. bifolia* (Table 4)*, in contrast with the results of *Tollsten & Bergström (1993)* where *P. chlorantha* contained a higher percentage of lilac aldehyde. Furthermore, among the ten volatile compounds identified, the relative amount of three compounds (ocimene, 1,2-hexanediol-2-benzoate and santolina triene) was emitted in high percentage only by *P. bifolia* and the intermediate morphotypes compared with the *P. chlorantha* population. Also other compounds were dominant, such as benzenoids in *P. bifolia* compared with *P. chlorantha* (Table 4), which was in accordance with the results of *Tollsten & Bergström (1993)*. By contrast, the floral compound 3,7-Dimethyl-1,3,6-octatriene is present in a high percentage in *P.bifolia* compared to the intermediate morphotypes and *P. chlorantha* species. This compound was observed to be pheromone involved in social regulation in a honeybee colony (*Maisonnasse et al., 2010*) but not directly involved in attraction of nocturnal moths (http://www.pherobase.com/database/compound/compounds-detail-cis-beta-ocimene.php).

MRPP analysis indicates that floral scents were significantly differentiated among groups in floral scent composition ($A = 0.551$, $\delta obs = 0.262$, $\delta exp = 0.583$, $P < 0.001$). The ANOSIM analysis was in accordance with the result of MRPP analysis, which showed a significant difference between *Platanthera* groups ($R = 0.748$, $P < 0.001$). Moreover, Tukey's Honest Significant Differences and post-hoc test revealed that there was a significant difference in the variance dispersion of floral scents between *P. bifolia* and *P. chlorantha* ($P = 0.002$) and between *P. chlorantha* and intermediate morphotypes ($P = 0.011$).

The analysis of overall floral odour similarity with HCAST produced a dendrogram which shows that investigated sympatric populations are resolved into two clusters that were supported statistically (AU values > 80%) (Fig. 9). The first cluster contained the sympatric *P. chlorantha* and the second two subclusters contained intermediate and *P. bifolia* individuals sympatric with them. The results of this analysis show a significant similarity of chemical patterns in floral scent composition with *P. bifolia* of all intermediate morphotypes sampled (Fig. 9).

## DISCUSSION

*Platanthera* individuals with intermediate shape column (gynostemium width/length) have been observed and described in several geographical areas (in Sweden, *Nilsson, 1983*; *Nilsson, 1985*; *Plepys, Ibarra & Lofstedt, 2002b*; *Maad & Nilsson, 2004*; in Netherlands, *Claessens & Kleynen, 2006*; *Durka et al., 2017*; in Austria, *Perko, 1997*; *Künkele & Baumann, 1998*; in Germany and Belgium, *Durka et al., 2017*). Exceptional situations with putative isolated hybrids or with a limited numbers of parental species have also been observed (*Perko, 1997*; *Perko, 2004*; *Claessens & Kleynen, 2006*; *Durka et al., 2017*).

Our morphological analyses confirmed the expectations with intermediate morphotypes diplaying significant differences in column morphology between both *Platanthera* species. We also observed that the allopatric population of *P. bifolia* exhibited more extremes values of floral column compared to the sympatric one. The multivariate analysis (PCA) showed that the analysed floral characters were exhaustively discriminant between the two *Platanthera* species, which were represented in two separate clusters, where intermediate morphotypes were closer to *P. bifolia* group (Fig. 3).

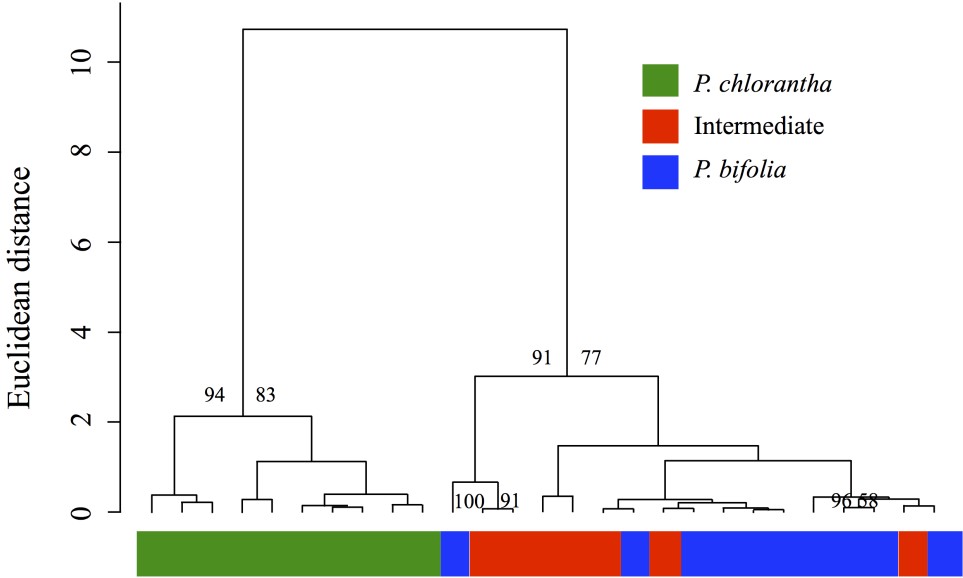

**Figure 9  Cluster dendrogram produced by hierarchical cluster analysis.** Cluster dendrogram produced by hierarchical cluster analysis with Ward's method using Euclidean distances among floral scent samples (relative proportions in % of the total blend) of the *Platanthera* sympatric groups investigated. Approximately Unbiased (AU) Ps > 80% are indicated above the branches of the dendrogram.

Our molecular investigations (AFLP) also revealed a good separation between the two *Platanthera* species (PCoA), but in this case, plants with intermediate floral morphological traits, could not be genetically separated from *P. bifolia* (full overlap of AFLP's profiles) (Fig. 5). Similarly, the analyses with STRUCTURE and NEWHYBRIDS revealed two distinct clusters, one containing *P. bifolia* and intermediate morphotypes and the other containing *P. chlorantha* species (Fig. 6).

Accordingly, the Hybrid Index analysis confirmed that intemediate morphotypes belong to *P. bifolia* group, showing an average value of hybrid index of 0.1 considering both sympatric sites. On the other hand, individuals firstly morphologically identified as *P. chlorantha* showed a hybrid index ranging between 0.4 and 0.7. This unexpected result could be probably due to the high genetic differentiation between allopatric *P. chlorantha* that we considered as reference species in INTROGRESS and the two sympatric populations as confirmed from the Fst values (Botton: 0.12; Bois Niau: 0.14). Another hypothesis has been proposed basing on the assumption of *Bateman, James & Rudall (2012)*, who explored both nuclear and plastid genomes in *P. bifolia* and *P. chlorantha*, by identifing only one variable site in the ITS region (not species-specific) that distinguished the two species. The evidence for *P. chlorantha* being derived relative to *P. bifolia* comes from phylogenetic comparison with other closely related species of *Platanthera*, i.e., those included in the ITS tree of *Bateman et al. (2009)*. Accordingly to this statement, the maximum 0.7 value of *P. chlorantha* obtained in INTROGRESS could reveal that most of *P. bifolia* genome is contained in *P. chlorantha* and the exclusive loci, since they are shared between the two species, would be few.

Moreover, molecular analyses also revealed that about 17% and 7% of all sampled individuals displayed an admixed gene pool, indicating that hybridization and introgression between the two taxa had occurred in both sympatric sites.

The few intermediate genotypes observed in the studied sympatric populations may be the result of an ancestral polymorphism that persists in the sister species (incomplete lineage sorting) or the result of an asymmetric gene flow, which may depend on the relative sizes of the two sympatric populations and of the various species of pollinating moths. It is already known that evolution may occur at the level of loci (*Wu, 2001*; *Nosil & Schluter, 2011*) and hybrid forms often represent a transitional phase in a much larger dynamic exchange of genetic material between parental lines, via backcrossing and introgression (*Gompert & Buerkle, 2010*; *McIntosh et al., 2014*). We can also hypothesize that hybridization between these two studied species is restricted to certain stochastic events.

Nevertheless, since most of these intermediate morphotypes unequivocally belonged to *P. bifolia* gene pool, the most likely scenario that could explain this situation might be that, within the genetic background of *P. bifolia*, recent selection acting on a genetic polymorphism would lead to a modification of the morphology of floral column. Thus, we supported the hypothesis of *Bateman, James & Rudall (2012)* according to which a gymnostemium widening could reflect a mutation of a very limited number of genes that are involved in the phenotypic shift from *P. bifolia* to *P. chlorantha*. We also suspected that this limited number of genes might be also responsible for the variation in gynostemium width in intermediate *P. bifolia* individuals. However, several examples of morphological versus molecular divergence have also been recorded in other European orchid clades (*Bateman et al., 2003*; *Pellegrino et al., 2005*; *Bateman et al., 2011*). One example could be represented by *Dactylorhiza incarnata* aggregate, which was rich in phenotypic diversity (*Bateman & Denholm, 1983*) but shows little or no variation in allozymes, ITS sequences, plastid haplotypes or even AFLPs (*Hedrén, Fay & Chase, 2001*), even though more recent studies by *Hedrén (2009)* have revealed greater, and more structured, molecular differences among ecotypes of *D. incarnata*.

Additionally, genetic compatibility between *P. bifolia* and *P. chlorantha* was found through manual interspecific crosses experiments, where the proportion of viable seeds was not different between the two species and not significantly higher compared to intraspecific crosses. The estimation of post-pollination pre and post-zygotic indices indicated that interspecific crosses performed better than intraspecific ones, suggesting that species were not completely isolated.

Furthermore, since the association with mycorrhizal fungi may contribute to maintaining a post-zygotic isolation barrier between the orchid species (*Jacquemyn et al., 2010*; *Jacquemyn et al., 2012b*; *Reinhart, Wilson & Rinella, 2012*; *Bateman et al., 2014*), in a previous study (*Esposito et al., 2016*), we showed that in our studied sympatric populations, mycorrhizal fungi were most likely not directly involved in maintaining species boundaries.

Overall, besides the substantial area of sympatry and the evident reproductive compatibility, we also observed that flowering phenology did not largely differ between the two species. The overlap in flowering phenology is high (RI close to zero), and it is unlikely to contribute to reproductive isolation between the two species. Moreover, the

flowering time of intermediate morphotypes preceded the one of the sympatric *P. bifolia*, (even if this difference was rather small) and follows the sympatric *P. chlorantha*. Despite the apparent lack of strong post-pollination barriers, remarkably similar distributions and a considerable overlap in ecological preferences, *Bateman, James & Rudall (2012)* considered surprising the fact that both *Platanthera* species did not co-occur more often as well as the significant paucity of confident records of hybrids between the two species.

Given all these considerations, we may speculate that the low rate of introgressed genotypes found in sympatry could be due to a combination of several pre-pollination isolation barriers, and their potentially complex interactions. In Orchidaceae, pre-pollination mechanisms seem generally to be particularly good at achieving isolation in sympatry despite often sharing pollinators (*Dressler, 1968*; *Cozzolino et al., 2005*; *Cozzolino & Scopece, 2008*), and these barriers have been often described, on average, to be twice as strong as post-pollination ones, by contributing more to total isolation (*Martin & Willis, 2007*; *Rieseberg & Willis, 2007*; *Lowry et al., 2008*; *Widmer, Lexer & Cozzolino, 2009*; *Baack et al., 2015*). For example, floral odour may represent a mechanism preventing that both species interbreed randomly, and as a pre-pollination barrier, may play a crucial role in upholding species isolation in sympatry (*Grant, 1949*; *Grant, 1994*).

Our chemical investigation of floral scent profiles in sympatry corroborated the genetic results on the belonging of those morphological intermediate individuals to *P. bifolia*. More precisely, it revealed that *P. chlorantha* group differed considerably in floral compounds from *P. bifolia*, whereas the intermediate morphotypes presented a similar chemical profile to *P. bifolia* (Table 4—except for 3,7-dimethyl-1, 3,6-octatriene which is in common with *P. chlorantha*). These findings of were broadly in agreement with previous studies on floral scent composition of *Platanthera* (*Nilsson, 1983*; *Nilsson, 1985*). Monoterpenes were, indeed, the most abundant compounds in the floral bouquets of all studied populations for *P. chlorantha* individuals; the floral scent was essentially composed of lilac aldehydes and alcohols. In contrast, a mixture of monoterpenes and aromatic esters was observed in *P. bifolia*. *Nilsson (1983)* suggested that the presence of lilac compounds in *P. chlorantha* could be an adaptation to noctuid moths, and aromatic esters in *P. bifolia* to sphingid moth pollination.

Also *Tollsten & Bergström (1993)* observed the divergence in floral scent composition between *Platanthera* species, by hypothesizing that floral scent might reflect variations in the local pollinator fauna (noctuid, geometrid, as well as sphingid moths), which may lead to an "ethological isolation". However, *Tollsten & Bergström (1993)* also pointed out that different populations of *P. bifolia* with contrasting pollinator spectra, actually possessed indistinguishable scent cocktails. Such an observation, of course, does not favour the idea that divergence of *P. chlorantha* with respect to *P. bifolia* would be driven by (subtle) divergence in scent blends.

Furthermore, the evolution of pre-pollination incompatibility among plants species (and populations) is thought to be associated with the specialized relationships with pollinators (*Stebbins, 1970*), and the divergence in column's morphology in *Platanthera* species may represent a significant mechanical barrier (*Nilsson, 1983*). Specializations to various methods of pollination by the same type of pollinator was already known in this

genus (*Efimov, 2011*). In *Platanthera* species, the morphological divergence, in particular the distance between the viscidia, determine the attachement of the pollinia on different parts of the pollinator's head (eyes or proboscis). The pre-pollination barrier due to pollinators may not be as hermetic as usually thought. Indeed, a noctuid species, namely *Cucullia umbratica*, was observed as a pollinator of intermediate individuals (*Claessens, Gravendeel & Kleynen, 2008*). The same species was also observed by us, visiting and pollinating intermediate as well as *P. bifolia* and *P. chlorantha* individuals, in the same populations as studied in this article (*Esposito, Merckx & Tyteca, 2017*).

However, some speculations have been made in order to explain the presence and the persistence of intermediate morphotypes in sympatry which displayed a morphologically hybrid's resemblance of column morphology and a genetical patterns shared with *P. bifolia* group. We may hypothesize, for instance, a system where the phenotype of a group remains unchanged, while selection seems to act only on the phenotype of the second group and bringing about a resemblance to the first group, by describing a typical scenario of advergent evolution (*Johnson, Alexandersson & Linder, 2003*). This system could be established in our study population, through which *P. bifolia* individuals tend to acquire *P. chlorantha*-like floral characteristics in sympatry. In plants, advergent evolution is primarily influenced by pollinators, which select flowers on the basis of their conditioned preferences (*Chittka & Thompson, 2001*).

Moreover, we may also speculate that among *P. bifolia* plants, the individuals tending towards *P. chlorantha'* s phenotype may be positively selected in order to attract and exploit the pollinators of this species. The analysis of female success, indeed, showed a significantly higher fruit set in *P. chlorantha* compared to *P. bifolia* in both sympatric sites (Fig. 4).

These results may also suggest a slightly higher fruit set of intermediate forms compared to *P. bifolia* group (Fig. 4B). Indeed, since reproductive success depends on the interaction between pollinators and column length, a better fit between them, may influence fruit rate through better pollinaria removal and deposition, and can shape the evolution of interspecific floral variation. Particularly, considering that pollinators penetrate the spur via its entrance, the distance separating the viscidia will be a crucial trait that dictate which potential pollinator will be the most efficient in transferring pollen among flowers, by influencing strongly the reproductive success. Moreover, in a previous study conducted by *Hapeman & Inoue (1997)*, the column morphology of *Platanthera* is considered evolutionary labile and easily shifted when subjected to pollinator-mediated selection.

Interestingly, the results of a study recently published by *Durka et al. (2017)*, who investigated the morphological and genetic variation between both *Platanthera* species and morphologically intermediate individuals (found isolated from parental species) in Germany, the Netherlands and Belgium, revealed the presence of three independent gene pools represented by *P. chlorantha*, *P. bifolia* and plants referred to as non-hybrid intermediates, which although phenotypically intermediate, were not of hybrid origin. Nevertheless, the existence of these intermediate plants as pure and autonomous populations genetically distinct could be explained by a genetic drift. By contrast, our observed populations of intermediate plants in Belgium are still mixed with *P. bifolia*, thus the reasons of their presence and persistence still remain an open question.

Thus, to provide better knowledge whether these sympatric *Platanthera* species just respond plastically to environmental conditions or are in a process of early speciation and specialization to local pollinators, further studies that will consider the evolutionary drivers of reproductive isolation and genomic basis of adaptive traits in natural populations, need also to be conducted.

## CONCLUSIONS

We may confirm that analyses based exclusively on morphological data are likely to fail to recognize hybrids accurately (*Rieseberg & Ellstrand, 1993*). The availability and the increasing ease of development of molecular markers have facilitated studies of potential cases of hybridization and introgression (*Rieseberg & Ellstrand, 1993*; *Martinsen et al., 2001*). A number of studies conducted on orchid species using molecular markers have confirmed the great utility of the latter (*Moccia, Widmer & Cozzolino, 2007*; *Pinheiro et al., 2010*; *Pinheiro et al., 2015*; *Pavarese et al., 2011*).

In this study, we investigated two sympatric contact zones between two closely related *Platanthera* species and we observed that individuals with intermediate morphology were genetically belonging to *P. bifolia* group. However, the assignment of these individuals as *P. bifolia* species has been more reliable only by providing a much more detailed picture of the genetic structure of a sympatric zone through the use of genome wide analysis.

Overall, we found a low rate of hybridization/introgression, together with an apparent lack of strong post-pollination isolation mechanisms, which allow us to speculate that it could be due to a combination of pre-pollination isolation barriers. Thus, it would be interesting to explore if variation in gynostemium morphology among species is the simple result of plasticity or may reflect adaptive evolution in response to pollinator-mediated selection.

This may be possible by conducting a selection study in sympatry in order to evaluate the presence of phenotypic selection acting on floral characters related to both sex functions in the context of different kinds of local pollinator's composition. These kinds of studies might provide a better framework for understanding patterns of pollinator-mediated selection or hypothetical advergent evolution in *Platanthera* sympatric species.

## ACKNOWLEDGEMENTS

We thank Thomas Merckx, Thomas Henneresse and Guy Deflandre for field assistance, Laurent Grumiau for lab assistance (chemical analysis) and Manon Martin for statistical support. We greatly appreciate the critical and constructive comments from Renate Wesselingh and Salvatore Cozzolino on earlier versions of the manuscript, as well as Richard Bateman, Barbara Gravendeel and anonymous reviewers for later versions. This is paper BRC 382 from the Biodiversity Research Centre of UCL.

# APPENDIX

Photographies deposited in the Herbarium of the Belgian National Botanic Garden, Meise (BR). In the author's number, "DT", Daniel Tyteca; "FE", Fabiana Esposito.

| Taxon | Locality | Collection date | Author's number | BR reference | Correspondence in article |
|---|---|---|---|---|---|
| *P. bifolia* | Navaugle | 1st July, 2012 | DT_0404 | **BR0000025789065V** | |
| | Navaugle | 1st July, 2012 | DT_0440 | **BR0000025789072V** | |
| | Navaugle | 8 July, 2013 | DT_0701 | BR0000025789089V | Fig. 1A |
| | Navaugle | 8 July, 2013 | DT_0703 | **BR0000025789096V** | |
| | Bois Niau | 3 June, 2011 | DT_0980 | **BR0000025789003V** | Fig. 1B |
| Intermediate | Bois Niau | 3 June, 2011 | DT_0974 | **BR0000025788990V** | |
| | Bois Niau | 21 June, 2010 | DT_0836 | **BR0000025789010V** | |
| | Bois Niau | 21 June, 2010 | DT_0840 | **BR0000025789027V** | |
| | Botton | 25 May, 2011 | DT_0947 | **BR0000025789034V** | Fig. 1C |
| | Botton | 25 May, 2011 | DT_0953 | **BR0000025789041V** | |
| | Botton | 25 May, 2011 | DT_0958 | **BR0000025789058V** | Fig. 1D |
| *P. chlorantha* | Bois Niau | 21 June, 2010 | DT_0850 | **BR0000025789102V** | Fig. 1E |
| | Botton | 18 May, 2007 | DT_0035 | **BR0000025789119V** | |
| | Transinne | 4 July, 2013 | FE_4196 | BR0000025789126V | Fig. 1F |
| | Transinne | 4 July, 2013 | FE_4222 | **BR0000025789133V** | |

## Funding

This study was supported by grants-in-aid from the Fonds Scientifique de Recherche of the Université catholique de Louvain for two years between 2013 and 2015. The funders had no role in study design, data collection and analysis, decision to publish, or preparation of the manuscript.

## Grant Disclosures

The following grant information was disclosed by the authors:
Fonds Scientifique de Recherche of the Université catholique de Louvain.

## Competing Interests

The authors declare there are no competing interests.

## Author Contributions

- Fabiana Esposito conceived and designed the experiments, performed the experiments, analyzed the data, wrote the paper, prepared figures and/or tables, reviewed drafts of the paper.
- Nicolas J. Vereecken conceived and designed the experiments, performed the experiments, analyzed the data, contributed reagents/materials/analysis tools, wrote the paper, prepared figures and/or tables, reviewed drafts of the paper.

- Maddalena Gammella analyzed the data, wrote the paper, reviewed drafts of the paper.
- Rosita Rinaldi and Pascal Laurent performed the experiments, analyzed the data, contributed reagents/materials/analysis tools.
- Daniel Tyteca conceived and designed the experiments, performed the experiments, wrote the paper, prepared figures and/or tables, reviewed drafts of the paper.

## Field Study Permissions

The following information was supplied relating to field study approvals (i.e., approving body and any reference numbers):

Field experiments were approved by the Département de la Nature et des Forêts of the Walloon Region of Belgium.

## Data Availability

The raw data is included in the Supplemental Files and in Tables 1–4 in the manuscript.

## Supplemental Information

Supplemental information for this article can be found online at http://dx.doi.org/10.7717/peerj.4256#supplemental-information.

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
