# Peer review of "Characterization of sympatric Platanthera bifolia and Platanthera chlorantha (Orchidaceae) populations with intermediate plants"

_PeerJ, doi:10.7717/peerj.4256_

## Round 0.1 · original submission · Minor Revisions

Dear Daniel et al.,

Please excuse the formality (and periodically poor English) of the above obligatory blurb. I should begin by apologising for the delay in providing reviews and editor's comments. This was my responsibility and reflects a deal that I struck with the people who I regarded as the most appropriate reviewers – both individuals requested extensions to the normal review period. This decision in turn led to friction between myself and PeerJ staff, which is likely to precipitate my resignation from PeerJ's editorial board once I have safely overseen final acceptance of your revised manuscript.

The reviews, once the second was finally obtained yesterday, are less detailed than I would have hoped, particularly given the time allocated. One reviewer has recommended acceptance with Major Revision, the other Acceptance with Minor Revision. I am inclined toward the view of the more generous reviewer, and so will be happy to accept your manuscript after some modest textual revisions. I will, however, make a few comments of my own, both in the context of, and in addition to, those comments offered by the two reviewers.

Reviewer 1 raises several criticisms of your study, but most boil down to a single underlying criticism – that you studied an insufficient number of populations, and that because of this, several aspects of your Discussion are "too speculative" and that you therefore need to perform "additional experiments in order to reach the scientific value [threshold of acceptability?–RB] of PeerJ." In theory, I agree with the sentiment behind the reviewer's second statement. However, in practice, I note that (1) the amount of work that can be performed during a particular study is heavily dependent on resources available, and (2) that the laudable threshold of acceptability for PeerJ is simply that the study should be methodologically sound. I am satisfied that this manuscript easily passes that basic criterion, not least because three appropriate analytical approaches have been combined in this study (two more than in many equivalent papers!) and subjected to admirable rigorous statistical analyses. And although I also agree that much of the Discussion is speculative, I am perfectly happy to accept speculation provided that it is clearly labelled as such (and, in this case, it is).

Reviewer 1 also raises concerns in two less crucial areas that I too had identified. Firstly, Reviewer 1 notes that P. bifolia also occurs in calcareous grasslands. Certainly, in Britain, P. bifolia occurs as two clearly distinct ecotypes that are unlikely to ever exchange genes – one occupying grasslands and woods on calcareous soils, the other heathlands on acidic soils. Thus far, the nature of these two ecotypes has not been subject to detailed investigation, but they have obvious relevance to the present study and do at least merit brief mention.

Secondly, Reviewer 1 notes that the dominant compounds in the scent data do not show intermediacy in the "intermediates". I agree with the reviewer that this fact merits brief discussion, not least because one obvious explanation would be over-expression of 1,2-Hexanediol-2-benzoate in a hybrid swarm!

Reviewer 2 offers two main criticisms, both of which I had in fact raised with the authors 16 months ago. Firstly, I agree with Reviewer 2 that it is desirable to include a more explicit statement of how class membership of each individual plant was decided (i.e. Taxon 1, Taxon 2, Intermediate). Secondly, and more interestingly, I agree with the reviewer that the (to me, fascinating) issue of parental bias should be addressed in the Discussion. I refer the authors to a paragraph of text that I sent to them 16 months ago:

"It might be argued that the much stronger similarity of the so-called intermediates to bifolia in practice precludes a hypothesis of gene-flow, but I and others have repeatedly shown that F1 hybrid orchids are rarely close to 50:50 intermediate in phenotype between their parents; some genes are over-expressed, other under-expressed, others yield phenotypes that favour one parent or the other, so that comparatively few phenotypic characters blend roughly equally properties of the two parents. This makes it especially difficult to identify hybrids through morphology when as few characters have been measured as here. In addition, I have speculated that there exists an epigenetic overprint that causes the ovule parent to contribute more to the F1 orchid than does the pollen parent. Thus, the logic apparently invoked by the present authors – that near-equal intermediacy indicates hybridisation, whereas intermediacy that is highly skewed toward one of the co-occurring bona fide species indicates advergence within a species – is insupportable."

In this context, I have through the years published three morphometric studies of natural hybrids of European orchids – two also involving DNA data – and I presently have a fourth such study in press (Bateman & Farrington, 1987, Watsonia 16: 397; Bateman & Hollingsworth, 2004, Taxon 53: 43; Bateman, Smith & Fay, 2008, Bot J Linn Soc 157: 687; Bateman, Murphy & Tattersall, 2017, New J Bot, in press). These studies reliably showed strong phenotypic bias toward the ovule-parent. In the case of the most recent study, morphometric analysis based on 30+ characters conclusively identified the hybrid as a member of the maternal species(!), whereas the DNA data clearly stipulated that the plant was in fact an F1 hybrid.

Two lesser points: Unlike Reviewer 2, I do not believe that the molecular dating of Inda et al. (2012) is relevant here (Line 464), as the level of molecular divergence identified by you is too little to succumb to analyses based in dichotomy when the processes in play are clearly reticulate. And unlike Reviewer 2, I would not argue that formal evidence of permission is needed for limited and non-destructive cross-pollination experiments to be performed in the field (Line 606) – in my personal opinion, this issue belongs very much in the collective consciences of the researchers in question. I can only state that I would not have sought formal permission under most such circumstances.

A few comparatively minor comments/criticisms of my own:

Lines 463–5: "given their great similarity of genotypes, Bateman et al. (2012) assumed that P. chlorantha would originate from within P. bifolia species."
Wrong! This was actually the evidence used by Bateman et al. to assert that chlorantha and bifolia are sister-species. Rather, the evidence for chlorantha being derived relative to bifolia was phylogenetic comparison with other closely related species of Platanthera, not least those included in the ITS tree of Bateman et al. (2009).

Lines 482–4: "Thus, we supported the hypothesis of Bateman et al. (2012) according to which an expanded gymnostemium could reflect a mutation of a very limited number of genes that are involved in the phenotypic shift from P. bifolia to P. chlorantha." And:
Lines 566–8: "in a previous study conducted by Hapeman & Inoue (1997), the column morphology of Platanthera is considered evolutionary labile and easily shifted when subjected to pollinator-mediated selection."
I would say that the phenotypic distinction could easily boil down to a single base in a single gene that either mutated or epimutated (e.g. via merhylation). As stated by Bateman et al. (2012), the distinction is that the stigmatic surface of chlorantha is over-developed relative to that of bifolia. And Hapeman & Inoue (1997) presented no evidence in support of their assertion. If control of gynostemium is as simple as I believe then this would facilitate the column diversity best summarised by Efimov (2011), but of course, it would also facilitate other groups of evolutionary mechanisms such as drift.

Lines 488–90: "One example could be represented by Dactylorhiza incarnata aggregate, which was rich in phenotypic diversity (Bateman & Denholm 1983) but shows little or no variation in allozymes, ITS sequences, plastid haplotypes or even AFLPs (Hedrén et al. 2001)."
I broadly agree, but must admit that more recent studies by Hedrén and his collaborators have revealed greater, and more structured, molecular differences among ecotypes of D. incarnata.

Lines 541–2: We have, indeed, evidence for rapid morphological evolution of Platanthera genus in connection with pollination shifts (Efimov 2011)."
In the absence of relevant fossils, only molecular data can give any hint (and even that dubious!) regarding the rapidity of evolution. Efinov's study was pursued entirely within the realm of typological morphology. And with the exception of chlorantha and bifolia, most Platanthera species are subtended by ITS branches of at least moderate length, in contrast with, for example, all Ophrys or Serapias species. There is nothing obviously recently radiative about the genus Platanthera as a whole.

Line 560–1: "These [seed set] results may also suggest a slightly reproductive advantage of intermediate forms compared to P. bifolia group (Figure 4B)."
As I would anticipate, the entire Discussion is couched within a traditional neoDarwinian framework. But do the co-authors, really, truly, hand on heart believe that the evolutionary success or failure of an orchid lineage rests with subtle differences in the average numbers of viable seed produced per individual? There are so many objections available to deploy against this touching belief, but here I'll settle for just two: orchid plants do not compete directly for anything, and I calculate that the average Gymnadenia conopsea plant (and, I would guess, Platanthera) produces about 220,000 viable seeds in its lifetime – only one of those seeds need reach reproduce maturity for the population size to be maintained. Gods may not play dice, but nor do orchids enter the Euro-Lottery.

Lastly, I would argue that the authors could usefully include a little more information from the – to some degree competing – study recently published by Durka et al. (2017). They might also wish to include the companion paper to Durka et al. recently published by Baum & Baum (2017, AHO 49: 133), where they formally (and controversially) establish their 'third Platanthera morph' under the epithet P. muelleri.

I will end by stating that I do not believe that it will take the authors long, or cause them much hardship, to make minor amendments to the text in the light of the comments from myself and the two referees. Indeed, with one arguable exception, none of the above criticisms identifies an unequivocal error in the submitted material, and I would be willing to accept the manuscript unchanged if the authors saw no value in first indulging in a further (modest) upgrade to their previously submitted text. Either way, please ensure that you first remove the final few linguistic weaknesses from the text, noting that PeerJ does not offer copy-editing support.

Richard Bateman (10.11.2017)

Reviewer 1 ·

Basic reporting

Authors need to improve the flow and the clarity of the text in introduction and discussion parts.

Literature is almost sufficient; information of the reproductive strategies of Platanthera in American (where there are more diverse) would be added.

Line 68. P. bifolia is also largely present in calcareous grasslands

Experimental design

Line 104: In the end of introduction, the main research question is curiously presented “In order to determine whether intermediates are indeed hybrids, ...”. What are the alternative hypotheses? This part needs to be better explained.

Why intermediate individuals are investigated in only two populations while mixed populations between these two species are frequently observed (as said line 71). Why scents are investigated in only one of these two mixed populations? And why scents are not investigated in two control population? Why autogamy is investigated in only one population and with so few number of individuals (see table 1)? Why morphology is investigate on the base of only four traits? Why authors did not justify the selection of these morphological traits? What is the interpretative value of the number of days of co-flowering between only two populations of two species and during only one year? All these points strongly reduce the explanative power of this study. This situation renders the results very poorly convincing. All interpretations at species level are then based to a comparison between individuals from only two (and sometimes one) populations!

Validity of the findings

See before. Method used explained that results are poorly convincing.

In the figure 2A, how authors explain that sympatric and allopatric populations of P. bifolia are significantly different?

In the table 4, authors have to add both the retention index for each compounds, and the occurrence of each compounds of each species (and repeated the number of sampled individuals per species). The last information about occurrence is a key indicator to determine the evolutionary stability of each compound, especially when the number of compounds is very low.

The sentence Lines 435-436 “The results of this analysis show a significant similarity of chemical patterns in floral scent composition with P. bifolia of all intermediate morphotypes sampled” is indeed in contrast with the opposite values of relative proportions for 3,7-Dimethyl-1,3,6-octatriene and 1,2-Hexanediol-2-benzoate indicated in table 4. How authors explain this contrast?

Several parts of discussion are too speculative, like the involvement of mimicry system (549) or genetic drift (L 575). For example, how mimicry can occur if morphology and scent are different?

Additional comments

The resuls presented here mainly concern one population (for scents and pollination experiment) and only two for morphology and AFLP analyses. The absence of pollinator observations is a pity. Thus the generalisation of these results is quite low. Several points may be improved by additionnal experiments in order to reach the scientific value of PeerJ.

·

Basic reporting

This is a generally well written manuscript on the genetic, chemical and morphological characterization of plants of the terrestrial orchid species Platanthera bifolia and P. chlorantha with intermediate morphologies occurring in Belgium to investigate the possibility that hybrids are evolving in mixed populations in response to local pollinator-selected selection.

Experimental design

My main concerns are with the fact that it is not clear whether the genotypic frequency classes were assumed or experimentally verified and that the maternal effect is not mentioned and discussed to explain the results obtained.

Validity of the findings

Adding more information about the assignment of the genotypic frequence classes and the maternal effect is recommended.

Additional comments

Abstract line 25: The sentence ‘The results of the genetic analyses showed the same genetic patterns of morphologically intermediate individuals with the P. bifolia group’ is inclear so I recommend rephrasing it into ‘Morphologically intermediate plants had similar genetic patterns as the P. bifolia group’.

Introduction line 68: What is meant with a ‘fresh’ meadow? Please rephrase.

Materials and Methods line 120: ‘… for their attachment to the base of the pollinator’s head’. Please rephrase into ’… to various organs on the pollinator’s head’ to make the text more compatible with the information provided two paragraphs further down on attachement of pollinia to either the proboscis or eyes.

Materials and Methods lines 156-158: Was this done noninvasively?

Materials and Methods lines 263-264: The authors state that they ‘used six genotypic frequency classes to classify the analyzed individuals: pure parental species, F1, F2, backcross to each parental species. It is unclear how the authors assigned these classes. Were F1, F2 and backcrosses made by the authors themselves? If not, the authors should make more explicit here that they assumed the individuals to belong to these classes but that this was not verified experimentally.

Results line 335: Please remove ‘the’ in ‘… traits explained the 89%...’

Results line 420: Please change ‘the honey bee colony’ into ‘a honey bee colony’.

Discussion line 439: In the sentence ‘… individuals with intermediate column…’ is not clear what exactly is intermediate here so please add a specification such as ‘shape’ or ‘size’ here.

Discussion line 464 and further: I agree that the hypothesis that P. chlorantha evolved from P. bifolia is plausible. The authors could provide a bit more information about the timeframe in which this happened, though, for instance by referring to Inda et al. (2012) Annals of Botany, in which a molecular clock analyses is presented for a.o. the divergence of P. chlorantha and P. bifolia. According to these analyses, this divergence started only quite recently (i.e. in the Pleistocene), which could explain the fact that lineage sorting is still ongoing.

Discussion line 483: What is meant with the word ‘expanded’? Elongation? Widening? The authors should make this more clear.

Discussion in general: What I missed is the possible influence of the so-called maternal effect in plants (Roach and Wulff, 1987 Annual Reviews), where the mother contributes most to the phenotype of the offspring. It might very well be that the plants with intermediate phenotypes are genetically, chemically and morphologically most similar to P. bifolia because this species received most of the pollinia from B. chlorantha! The authors should mention this and the possibility to verify this by carrying out man-made crosses and investigating the resulting phenotypes.

Discussion line 571: Please add that this particular study was carried out ‘in Austria’ as most readers will not know this and assume that Durka et al. (2017) also studied plants in Belgium. Likewise, please add ‘in Belgium’ to the sentence ‘… our observed populations of intermediate plants…’

Acknowledgements line 606: please provide information on permits obtained for the experimental crosses by hand pollination carried out in the field to determine the level of compatibility between the species investigated.

---

## Round 0.2 · accepted · Accept

Dear All,

It is my pleasure to accept your (pre-Christmas) resubmission for publication in PeerJ – I look forward o seeing it in print (my guess is that it will be published toward the end of January).

With best wishes: Richard